# Land use and semen quality: A fertility center cohort study

**Seung-Ah Choe**[1], **Seulgi Kim**[2], **Changmin Im**[3], **Sun-Young Kim**[4], **Gregory Wellenius**[5], **You Shin Kim**[6], **Tae Ki Yoon**[6], **Dae Keun Kim**[7]*

1 Department of Preventive Medicine, Korea University Medical College, Seoul, South Korea, 2 Graduate School of Public Health, Seoul National University, Seoul, South Korea, 3 Department of Geography, Korea University, Seoul, South Korea, 4 Department of Cancer Control and Population Health, Graduate School of Cancer Science and Policy, National Cancer Center, Goyang-si, Gyunggi-do, South Korea, 5 Department of Environmental Health, Boston University School of Public Health, Boston, MA, United States of America, 6 Department of Obstetrics and Gynecology, CHA Fertility Center Seoul Station, CHA University School of Medicine, Seoul, South Korea, 7 Department of Urology, CHA Fertility Center Seoul Station, CHA University School of Medicine, Seoul, South Korea

* dkkim@cha.ac.kr

## Abstract

This study explored the association between built environment and semen parameters among men who sought fertility evaluation. We used a data of 5,886 men living in the Seoul capital area whose semen was tested at a single fertility center during 2016–2018. Distance to fresh water, the coast, major roadways, and neighborhood greenness measured by Normalized Difference Vegetation Index (NDVI) were evaluated. Outcome indicators were semen volume, sperm concentration, percentage of progressive motility, vitality, normal morphology, and total motile sperm count. Linear regression models were fitted to standardized values of six semen indicators. Majority of men were white-collar, clerical, and service workers. Linear associations between built environment features and semen quality indicators were not evident except for NDVI within 500 m and sperm vitality (β = 0.05; 95% confidence interval (CI): 0.01, 0.09). The 2nd quartile of distance to fresh water was associated with lower progressive motility compared to the 1st quartile (β = −0.10; 95% CI: −0.17, −0.03). Proportion of vitality was higher among men in the 2nd quartile of distance to roadways than those in the 1st quartile (0.08; 95% CI: 0.01, 0.15). Men in the 2nd quartile of NDVI had higher total motile sperm count (0.09; 95% CI: 0.01, 0.17). In the multi-exposure model, the positive association between NDVI and vitality remained (0.03; 95% CI: 0.00, 0.06). We observed potential evidence regarding the impact of built environment on male fertility, specifically a positive association between residential greenness and sperm vitality among men with a history of infertility.

## Introduction

Growing evidence suggests potential impacts of the outdoor environment on human health. It has been suggested that components of built and the natural environment may influence levels

(seungah@korea.ac.kr) or corresponding author (kdg070723@gmail.com) on reasonable request. Interested researchers may also contact GCI2840@chamc.co.kr, Institutional Review Board of Gangnam CHA Hospital for data access.

**Funding:** This research was supported by the National Research Foundation of Korea grant (NRF-2016R1D1A1B03933410 and 2018R1D1A1B07048821, https://www.nrf.re.kr), which is funded by the Korean Government (recipient: SAC). The funders had no role in study design, data collection and analysis, decision to publish, or preparation of the manuscript.

**Competing interests:** The authors have declared that no competing interests exist.

of psychological stress, physical activity, and social relationships; and thereby, potentially improve or worsen human health and wellbeing [1–3]. For example, neighborhood green space has been associated with many beneficial health effects, including reduced all-cause and cardiovascular mortality and improved mental health, possibly mediated by less air pollution, heat and stress, and increased physical activity and social contacts [4].

Male reproductive function is highly sensitive to various physical agents generated by industrial activities [5, 6]. Prior studies revealed the association of semen quality with air pollution [7, 8], heat [9, 10] and pesticides [11]. In addition, semen quality itself reflects general health condition, since it is affected during the early stage of medical disorders [12, 13]. Therefore, assessing the relationship between residential environment and semen quality would expand our understanding of the potential role of environmental factors in human reproductive health. Prior studies found that exposure to ubiquitous chemicals including endocrine disruptive chemicals and air pollutants is associated with reduced semen quality [14–16]. Given the association of physical environment with human fertility, male reproductive potential represented by semen quality may be associated with features of the built environment. In this study, we aimed to assess the association between the residential built environment and six parameters indicative of semen quality among men with a history of infertility.

## Methods

### Data

This study was a cross-sectional study conducted among men who undertook semen tests between January 2016 and September 2018 at the CHA Fertility Center Seoul Station, the largest single IVF center in the Republic of Korea. The study design was approved by the institutional review board of Gangnam CHA hospital (GCI-18-48). As a research involving the retrospective review, this study qualified a waiver of informed consent. This study extended the study population of our prior study of Seoul residents [2] to those living in Seoul capital area (Seoul, Incheon, and Gyeonggi provinces). Semen tests were conducted as an initial evaluation in all couples who visited the center for a diagnostic purpose. Eligibility criteria include being aged 20–69 years. The addresses of our study population were the Seoul capital area where the traveling time to the Seoul clinic is within one hour. Excluding those diagnosed with varicocele, azoospermia, cryptorchidism, and a known chromosomal abnormality, we obtained semen analysis results of a total of 5,886 Korean men. We included only the first examination result of each patient to minimize the possible impact of medical intervention. Information on body mass index (BMI), occupation, and smoking was retrieved from their medical records.

### Measurement of built environment

The Korean peninsula is mainly mountainous along its east coast, most of its river water flows west, and highly populated towns are located mostly in the north-west region. Four built environment components commonly used in prior studies were measured: distance to fresh water, distance to the coast, distance to major roadways, and Normalized Difference Vegetation Index (NDVI) [17–20]. We used distance to the nearest major roadway since it is often used as a proxy for long-term residential levels of air and/or noise pollution due to traffic [21]. Distance to the nearest fresh water body, coast, and the average NDVI within a 500 m circular buffer were assessed as indicators of neighborhood restorative environment, as in previous studies [20, 22]. The distance from the geocoded address to the environmental variables was calculated using ArcMap's Spatial Join analytical tool, which analyzes the spatial relationship between two geographical features. We defined the distance between any two features as the

shortest separation between them, such that the two features are closest to each other. Euclidean distance to environmental features was calculated up to the boundary of a polygon, not to the center or centroid. For creating NDVI map and geospatial analyses, we used ArcGIS Desktop v. 10.5 (ESRI, Redlands, CA).

River and lake data were integrated into data on fresh water. Both data sets were retrieved from the National Spatial Data Infrastructure (http://www.nsdi.go.kr). River data was retrieved on January 21, 2016, and lake data on July 5, 2019. Integrated data is a nationwide polygonal data set consisting of 209,216 inland water bodies. We used coastline data provided by the National Geographic Information Institute and retrieved via the National Spatial Data Infrastructure portal (www.nsdi.go.kr). The nationwide polyline dataset was compiled on July 5, 2019. We calculated the distance perpendicular to the closest coastline from a geocoded point.

Data on major roads were obtained from national standard node links provided by the Korean Transport Database (KTDB) of the Korea Transport Institute (http://www.ktdb.go.kr). The original road data set was compiled on September 20, 2019, and was classified into nine categories: national highways, metropolitan city highways, general national roads, metropolitan city roads, government-financed provincial roads, provincial roads, district roads, highway link lamps, and other roads. In this study, we defined major roads as national highways, metropolitan city highways, metropolitan city roads, highway link lamps, and roads more than six lanes wide in other classes.

For data on NDVI, we used Landsat 7 satellite data provided by the United States Geological Survey (USGS) (https://earthexplorer.usgs.gov/). We assessed NDVI over the entire satellite image of the Korean peninsula from a combination of 13 Landsat satellite images taken over June, September, and October 2017 for cloud-free observation. The reasons for combining satellite image data for the above three months are as follows: 1) Since the revisit time of Landsat 7 is 16 days, it takes at least three visits and a month and a half to cover the entire Korean peninsula; and 2) In order to improve the accuracy of the NDVI value, only satellite images with less than 10% of the area obscured by clouds during this period were extracted.

### Semen collection and assessment

Semen analysis was done as described in a previous study [23]. In brief, patients were asked to produce semen samples in the andrology laboratory by masturbating into a sterile plastic cup after 3 to 5 days of sexual abstinence. The semen specimen was left for 30 minutes at room temperature (22˚C–24˚C) for liquefaction. General semen quality parameters were assessed based on the 2010 World Health Organization (WHO) criteria [24]. Sperm morphology was analyzed after centrifugation of semen with a resuspended pellet dyed with Diff-Quik fixative solution. The fixed specimen was then immersed in oil dropped on a microscope slide and observed using x1000 polarized microscopy. We assessed six continuous indicators (volume, sperm concentration, percentage (%) of progressive motility, vitality, normal morphology, and total motile sperm count) obtained via semen analysis. Total motile sperm count is defined as the number of moving sperm in the entire ejaculate, and is calculated by multiplying the volume by the concentration (million/mL) by the motility (%).

### Statistical analyses

Descriptive analyses involved calculation of mean and standard deviations or frequencies and percentages (%) for demographic characteristics and semen quality parameters. We conducted multiple imputation by chained equations (MICE) for the missing covariate data [25], assuming data were missing at random and were conditioned upon the variables included in the imputation model. This study conducted three main analyses: First, we explored the pairwise

correlation structure between three built environment components and sperm quality indicators, which are standardized using z-scores. Second, we tested for heterogeneity and linear trends in the mean values of sperm quality indicators across quartiles of environmental exposures using the Kruskal-Wallis rank sum test and Kendall's rank correlation test, respectively. Third, after examining the shape of relationships using a generalized additive model with a spline function and adjustment for potential covariates (R software ver. 3.6.2), we used linear regression models to estimate the change in mean values of sperm quality indicators per interquartile range (IQR)-increase and for each quartile of exposure to the built environment (denoted as Q1, Q2, Q3, and Q4). We included individual characteristics such as age (categorized as 20s, 30s, 40s and 50s), BMI ($< 23$, 23–24.99, 25–29.99, and $\geq 30$ kg/m$^2$) based on the criteria for Asian populations [26], occupation (2 groups), current smoking (yes or no), season (Mar-May, Jun-Aug, Sep-Nov, Dec-Feb), and clustering effect of district ('*gu*', n = 68) in a generalized estimating equation to adjust for potential confounding effects. We did not include air pollution and regional deprivation index because the analytic model of this study included administrative district of home address, which is also the basis of estimation of exposure to air pollution [17, 27], noise [28], and deprivation index [29]. In addition, residential proximity to major roadways can be used as a proxy for exposure to traffic-related air pollutants, noise, and other spatial characteristics [30–32]. We additionally explored linear associations between built environment and six semen quality indicators with a multi-exposure model. A two-sided p-value of $< 0.05$ was considered statistically significant.

## Results

The mean age of the study population was 39 years (Table 1). The vast majority (96%) were white-collar workers, clerks, or service workers. Half of the men (49.3%) were obese (BMI $\geq 25$ kg/m$^2$) and were smokers at the time of examination. Regarding environmental exposures, the median distance to fresh water, the coast, and a major roadway was 382.8, 24869.5 and 486.7 m, respectively. The median NDVI was −0.2. The mean semen volume and concentration were 3.1 mL and 104.3 million/mL, respectively. The proportion of progressive motility and vitality were 45.6% and 62.6%, on average. The mean percentage of normal morphology was 3.7%. The mean value of the calculated total motile sperm count was 142.5 million per ejaculate. The pairwise correlation coefficients between four components of built environment and six sperm quality indicators were mostly low (S1 Fig). There was a positive correlation between the proportion of progressive motility and the proportion of vitality ($\rho = 0.74$). There were weak correlations among the four built environment components.

The mean value of progressive sperm motility was different across quartiles of distance to fresh water and a major roadway (S1 Table). Proportion of progressive motility was highest in those with 1$^{st}$ quartile of distance to fresh water. For distance to a major roadway, progressive motility was highest in the 2$^{nd}$ quartile and lowest in the 4$^{th}$ quartile. None of the semen quality parameters showed a linear trend across quartiles of built environment components.

Linear associations between built environment features and semen quality indicators were not evident except for NDVI within 500 m (Table 2). An IQR-increase in NDVI (0.1) was associated with 0.05-increase in z-score of vitality (95% confidence interval (CI): 0.01, 0.09). In the analyses using quartiles of exposures, living at the maximum distance to fresh water (i.e., in the 4$^{th}$ quartile) was generally associated with lower semen quality, but this did not reach statistical significance. The 2$^{nd}$ quartile of distance to fresh water (209.9–382.8m) was associated with lower percentage of progressive motility compared with the 1$^{st}$ quartile ($\beta = -0.10$, 95% CI: −0.17, −0.03). The proportion of vitality was higher in men in the 2$^{nd}$ quartile of distance to a major roadway compared with those in the 1$^{st}$ quartile (0.08; 95% CI: 0.01, 0.15). The association

**Table 1. Characteristics of 5,886 Korean infertile men.**

| Variables | Study population |
|---|---|
| Age (years) | 39.0 ± 4.6 |
| Body mass index (kg/m$^2$) | |
| < 23 | 1454 (24.7%) |
| 23–24.9 | 1530 (26.0%) |
| 25–29.9 | 2379 (40.4%) |
| ≥30 | 523 (8.9%) |
| Occupation | |
| White-collar workers, Clerks, Service workers | 5667 (96.3%) |
| Others | 219 (3.7%) |
| Current smoking | 3012 (51.2%) |
| Season | |
| Mar-May | 1652 (28.1%) |
| Jun-Aug | 1550 (26.3%) |
| Sep-Nov | 1169 (19.9%) |
| Dec-Feb | 1515 (25.7%) |
| Environmental exposures | |
| Distance to fresh water (m) | 453 ± 304 (Median: 383, IQR: 405) |
| Distance to coast (m) | 24870 ± 8210 (Median: 24610, IQR: 8948) |
| Distance to major road (m) | 1054 ± 1947 (Median: 487, IQR: 810) |
| NDVI | -0.1 ± 0.1 (Median: -0.2, IQR: 0.1) |
| Sperm parameters | |
| Volume (mL) | 3.1 ± 1.8 |
| Count (million/mL) | 104.3 ± 68.6 |
| Progressive motility (%) | 45.6 ± 13.2 |
| Vitality (%) | 62.6 ± 12.5 |
| Morphology (%) | 3.7 ± 1.8 |
| Total motile sperm count (million) | 142.5 ± 111.0 |

NDVI, Normalized Difference Vegetation Index; IQR, interquartile range. Continuous variables are presented as mean ± standard deviation. Others in occupation includes those unemployed. There was no missing case for age, smoking, BMI, occupation, and home address which are a part of medical record. For sperm quality indicators, there were missing cases in volume (n = 7), count (n = 46), progressive motility (n = 69), normal morphology (n = 80) and total motile sperm count (n = 53). No missing case was observed for vitality.

between NDVI and sperm vitality was positive when comparing the 4th (−0.08–0.35) versus 1st quartile (−0.34 −−0.20). Men in the 2nd quartile of NDVI had a higher total motile sperm count than those in the 1st quartile (0.09; 95% CI: 0.01, 0.17). The association between NDVI and the z-score for sperm vitality had the form of a cubic (S-shaped) pattern (S2 Fig). None of the semen quality indicators was associated with distance to the coast. In the multi-exposure model, linear associations between built environment features and semen quality indicators were not evident except for that between NDVI and vitality (0.03; 95% CI: 0.00, 0.06; Fig 1).

## Discussion

We did not find a consistent association between built environment and semen quality among men with a history of infertility. In single- and multi-exposure model, we observed that a higher value for NDVI within 500 m was positively associated with percentage of sperm vitality. The observed associations between environmental components and semen quality

**Table 2. Association between six semen quality parameters and four built environment components in single exposure models among 5,886 Korean infertile men.**

| | Semen volume | Sperm concentration | % of progressive motility | % of vitality | % of morphology | Total motile sperm count |
|---|---|---|---|---|---|---|
| Distance to fresh water | | | | | | |
| Per IQR increase | -0.01 (-0.04, 0.03) | 0.00 (-0.04, 0.03) | 0.01 (-0.03, 0.04) | 0.02 (-0.01, 0.06) | 0.00 (-0.04, 0.03) | 0.00 (-0.04, 0.03) |
| Quartiles of distance | | | | | | |
| Q1 | 0.00 (reference) | 0.00 (reference) | 0.00 (reference) | 0.00 (reference) | 0.00 (reference) | 0.00 (reference) |
| Q2 | -0.01 (-0.09, 0.06) | -0.03 (-0.10, 0.04) | **-0.10 (-0.17, -0.03)** | -0.02 (-0.09, 0.05) | -0.02 (-0.09, 0.05) | -0.06 (-0.13, 0.02) |
| Q3 | -0.03 (-0.11, 0.04) | 0.01 (-0.06, 0.08) | -0.07 (-0.14, 0.00) | 0.00 (-0.07, 0.07) | 0.00 (-0.07, 0.07) | -0.04 (-0.11, 0.03) |
| Q4 | -0.01 (-0.09, 0.06) | -0.03 (-0.10, 0.04) | -0.03 (-0.10, 0.04) | 0.02 (-0.06, 0.09) | -0.03 (-0.10, 0.04) | -0.03 (-0.11, 0.04) |
| Distance to coast | | | | | | |
| Per IQR increase | 0.01 (-0.02, 0.04) | 0.00 (-0.03, 0.03) | -0.01 (-0.04, 0.02) | -0.02 (-0.04, 0.01) | 0.00 (-0.03, 0.03) | 0.00 (-0.03, 0.03) |
| Quartiles of distance | | | | | | |
| Q1 | 0.00 (reference) | 0.00 (reference) | 0.00 (reference) | 0.00 (reference) | 0.00 (reference) | 0.00 (reference) |
| Q2 | -0.04 (-0.12, 0.03) | 0.03 (-0.05, 0.10) | -0.05 (-0.12, 0.03) | -0.02 (-0.10, 0.05) | 0.01 (-0.06, 0.09) | -0.02 (-0.09, 0.06) |
| Q3 | -0.02 (-0.10, 0.06) | 0.04 (-0.04, 0.11) | 0.00 (-0.08, 0.07) | -0.02 (-0.09, 0.06) | 0.03 (-0.05, 0.10) | 0.01 (-0.06, 0.09) |
| Q4 | 0.01 (-0.07, 0.09) | 0.04 (-0.04, 0.11) | -0.01 (-0.09, 0.06) | -0.04 (-0.11, 0.04) | 0.01 (-0.06, 0.09) | 0.04 (-0.04, 0.11) |
| Distance to major road | | | | | | |
| Per IQR increase | -0.01 (-0.03, 0.01) | 0.01 (-0.01, 0.03) | 0.00 (-0.02, 0.02) | 0.00 (-0.02, 0.02) | 0.00 (-0.02, 0.02) | 0.00 (-0.02, 0.02) |
| Quartiles of distance | | | | | | |
| Q1 | 0.00 (reference) | 0.00 (reference) | 0.00 (reference) | 0.00 (reference) | 0.00 (reference) | 0.00 (reference) |
| Q2 | -0.04 (-0.11, 0.04) | -0.03 (0.04, -0.1) | 0.04 (-0.03, 0.11) | **0.08 (0.01, 0.15)** | 0.06 (-0.02, 0.13) | -0.03 (-0.11, 0.04) |
| Q3 | -0.02 (-0.1, 0.05) | -0.04 (0.03, -0.12) | 0.00 (-0.07, 0.07) | 0.02 (-0.05, 0.09) | 0.04 (-0.03, 0.11) | -0.04 (-0.12, 0.03) |
| Q4 | -0.01 (-0.11, 0.1) | -0.03 (0.07, -0.13) | -0.08 (-0.18, 0.02) | -0.01 (-0.11, 0.09) | 0.05 (-0.05, 0.15) | -0.07 (-0.18, 0.03) |
| NDVI within 500m | | | | | | |
| Per IQR increase | 0.00 (-0.04, 0.04) | 0.01 (-0.03, 0.05) | 0.02 (-0.02, 0.06) | **0.05 (0.01, 0.09)** | -0.02 (-0.06, 0.02) | 0.00 (-0.04, 0.04) |
| Quartiles of NDVI | | | | | | |
| Q1 | 0.00 (reference) | 0.00 (reference) | 0.00 (reference) | 0.00 (reference) | 0.00 (reference) | 0.00 (reference) |
| Q2 | 0.03 (-0.05, 0.10) | 0.04 (-0.03, 0.11) | 0.06 (-0.01, 0.13) | 0.06 (-0.01, 0.13) | -0.04 (-0.11, 0.03) | **0.10 (0.03, 0.17)** |
| Q3 | 0.00 (-0.08, 0.07) | 0.00 (-0.07, 0.07) | 0.03 (-0.04, 0.10) | 0.05 (-0.02, 0.12) | -0.01 (-0.08, 0.06) | 0.05 (-0.02, 0.13) |
| Q4 | 0.03 (-0.06, 0.12) | -0.04 (-0.12, 0.04) | 0.05 (-0.03, 0.14) | **0.09 (0.01, 0.17)** | -0.03 (-0.11, 0.05) | 0.05 (-0.03, 0.14) |

IQR, interquartile range; NDVI, Normalized Difference Vegetation Index. Coefficients are calculated for standardized semen parameters (z-scores). Single exposure linear regression models included age, body mass index, occupation, smoking, season of semen test, and administrative district of home address. Results with P value <0.05 were bolded.

indicators were generally non-linear. For example, distance to fresh water was associated with lower percentage of progressive motility upon comparison of the first two quartiles. The 2nd quartile of NDVI was associated with higher total motile sperm count compared to the 1st quartile. To the best of our knowledge, this is the first report to assess the association between land use and semen quality using large hospital-based data.

Several components of physical environment have been known to be associated with male infertility. Heat exposure and extreme ambient temperature is associated with lower semen quality [17, 33]. Exposure to environmental noise is expected to be high when living close to a major road, and is associated with higher risk of subfecundity [34] or male infertility [35]. Specifically, environmental noise at the nighttime is associated with higher risk of oligozoospermia [28]. Higher air pollution is also reported to be related to semen abnormality [15, 36]. Although the strength of association with exposures is heterogenous across different semen indicators, the results of this study suggest a potential impact of the neighborhood's physical environment.

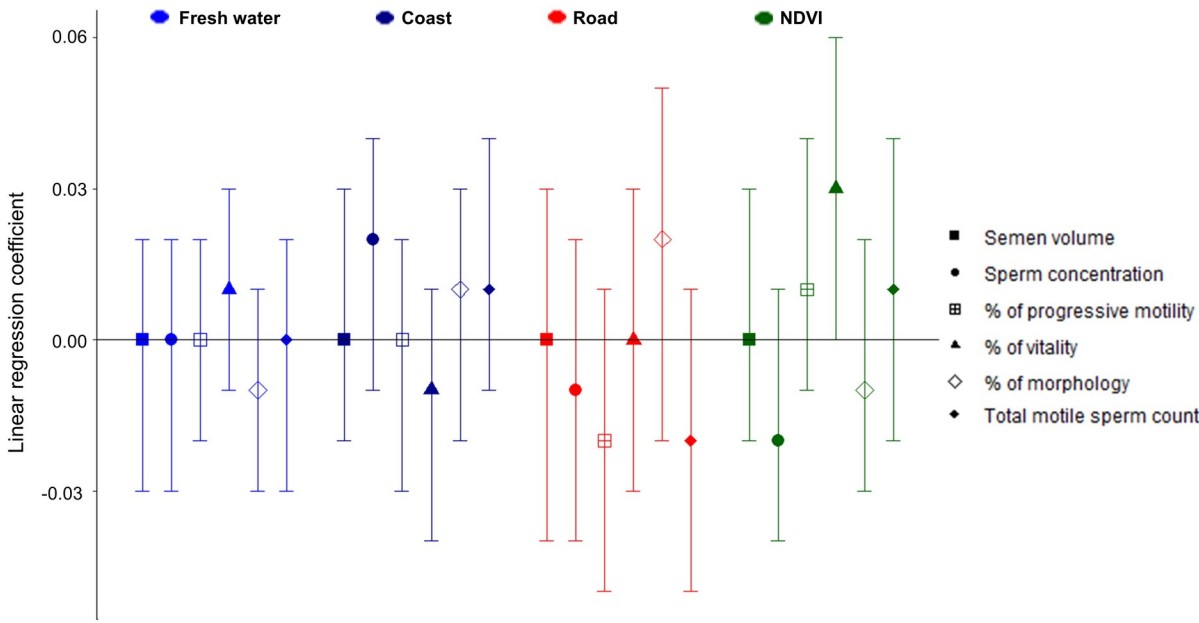

**Fig 1. Association between six standardized semen parameters (z-scores) and built environment components in a multi-exposure model among 5,886 Korean infertile men.** NDVI, Normalized Difference Vegetation Index. All coefficients are per interquartile range-increase of built environment components. Coefficients are calculated using a multivariable linear regression model including four built environmental components, age, body mass index, occupation, smoking, season of semen test, and administrative district of home address.

Our study found a heterogenous association between the built environment and semen quality across different exposures and semen indicators. There is limited knowledge regarding how each component of semen quality indicators is affected by different environmental exposures. Several studies have demonstrated increased sperm motility in physically active men [37, 38]. The observed inverse association between distance to fresh water and sperm motility would be explained by the contribution of neighborhood water body to ambient temperature [39]. Proximity to major roadways can be translated to higher exposure to environmental noise and traffic-related air pollution [30–32]. Similarly, residential greenness provided surface cooling reducing the potential impact of ambient heat on sperm quality [33, 40]. The non-linear association of proximity to fresh water or roadway with semen quality in our study may be explained by U-shaped association between built environment and active traveling [41]. Although our study did not detect a dose-response relationship, some of our results suggest the existence of a positive association of proximity to fresh water with sperm motility and of remoteness to roadway and NDVI with vitality.

The results of this study need to be interpreted with caution. First, findings of this study would have limited generalizability because the data is from single fertility center. Indeed, our study population was mostly restricted to white collar workers living in an urban area. We believe our study may still have important implications due to the use of hospital data belonging to a large infertile population who is expected to be particularly vulnerable to environmental exposure. Second, misclassification of exposure may have potentially occurred due to the use of residential address for exposure assessment, or due to the distance between the home address and the workplace, where patients may have spent a substantial amount of time. Assuming that the misclassification was non-differential, it may have biased our results towards the null [42]. There might have been hazardous effect by water pollution in some areas. According to the study of Mainali et al., with temporal and spatial variation in different

months of each season, water quality of the large river basin of Seoul metropolitan area exceeded 'poor' category up to 15 percent of times between 2012 and 2016 [43]. Given the relatively small proportion of water pollution in the area, we believe that the potential misclassification (benefit of proximity to fresh water) would have minimally biased the result. Lastly, there can be possible residual confounding effects caused by the lack of information of abstinence time, history of having any biological offspring, socioeconomic status including educational level, amount of smoking, and alcohol consumption. Future studies would need to minimize this potential bias by including these data.

## Conclusions

We did not find a consistent association between the built environment and different measures of semen quality among men with a history of infertility, although some features of neighborhood land use may be associated with semen quality, highlighting the potential impact of the built environment on human fertility. Further studies in different populations are required to add to the evidence on the impact of built environment on human reproduction and health.

## Supporting information

**S1 Fig. Correlation structure between built environment and sperm quality indicators.**
(DOCX)

**S2 Fig. Association between NDVI and sperm vitality in generalized additive model.**
(DOCX)

**S1 Table. Semen quality indicators in each quartile of distance to fresh water, coast, and road with NDVI.** Q1, lowest quartile; Q2, second quartile; Q3, third quartile; Q4, fourth quartile; NDVI, Normalized Difference Vegetation Index; $P^h$, P value for heterogeneity; $P^t$, P value for linear trend. Heterogeneity across quartiles was tested using Kruskal-Wallis rank sum test. Trend test was done with Kendall's rank correlation test. Results with P value $< 0.05$ were bolded.
(DOCX)

## Acknowledgments

We thank Sun-Wha Shim and Shi-Ha Jung for their technical support.

## Author Contributions

**Conceptualization:** Seung-Ah Choe.

**Data curation:** You Shin Kim, Dae Keun Kim.

**Formal analysis:** Changmin Im.

**Funding acquisition:** Seung-Ah Choe.

**Methodology:** Seulgi Kim, Changmin Im, Sun-Young Kim.

**Resources:** Seulgi Kim, Tae Ki Yoon, Dae Keun Kim.

**Software:** Changmin Im.

**Supervision:** Gregory Wellenius, Tae Ki Yoon.

**Visualization:** Changmin Im.

**Writing – original draft:** Seung-Ah Choe.

**Writing – review & editing:** Sun-Young Kim, Gregory Wellenius, You Shin Kim, Dae Keun Kim.

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
