## [Decision Letter · Decision Letter 0]

17 Jun 2021

PONE-D-21-13270

Land use and semen quality: a fertility center cohort study

PLOS ONE

Dear Dr. Kim,

Thank you for submitting your manuscript to PLOS ONE. After careful consideration, we feel that it has merit but does not fully meet PLOS ONE’s publication criteria as it currently stands. Therefore, we invite you to submit a revised version of the manuscript that addresses the points raised during the review process.

This is an interesting report which may deserve publication when a few concerns are addressed. The reviewers poihts should be followed point by point and critical issues need to be addressed during revision.

We look forward to receiving your revised manuscript.

Kind regards,

Stefan Schlatt

Academic Editor

PLOS ONE

Journal Requirements:

3. We note that Supplemental Figure 1 in your submission contain map images which may be copyrighted. All PLOS content is published under the Creative Commons Attribution License (CC BY 4.0), which means that the manuscript, images, and Supporting Information files will be freely available online, and any third party is permitted to access, download, copy, distribute, and use these materials in any way, even commercially, with proper attribution. For these reasons, we cannot publish previously copyrighted maps or satellite images created using proprietary data, such as Google software (Google Maps, Street View, and Earth). For more information, see our copyright guidelines: http://journals.plos.org/plosone/s/licenses-and-copyright.

3.1.    You may seek permission from the original copyright holder of Supplemental Figure 1 to publish the content specifically under the CC BY 4.0 license. 

3.2.    If you are unable to obtain permission from the original copyright holder to publish these figures under the CC BY 4.0 license or if the copyright holder’s requirements are incompatible with the CC BY 4.0 license, please either i) remove the figure or ii) supply a replacement figure that complies with the CC BY 4.0 license. Please check copyright information on all replacement figures and update the figure caption with source information. If applicable, please specify in the figure caption text when a figure is similar but not identical to the original image and is therefore for illustrative purposes only.

Reviewers' comments:

Reviewer's Responses to Questions

**Comments to the Author**

1. Is the manuscript technically sound, and do the data support the conclusions?

Reviewer #1: Yes

2. Has the statistical analysis been performed appropriately and rigorously? 

Reviewer #1: I Don't Know

3. Have the authors made all data underlying the findings in their manuscript fully available?

Reviewer #1: Yes

4. Is the manuscript presented in an intelligible fashion and written in standard English?

Reviewer #1: Yes

5. Review Comments to the Author

Reviewer #1: The authors report on 5886 men presenting for infertility evaluation in whom various markers of the ‘built-up environment’ were assessed for their impact on semen quality. Specifically, the distance to fresh-water, to the coast and to major roadways, and the neighbourhood ‘greenness index’, described as the Normalized Difference Vegetation Index [NDVI] as assessed from satellite data. A single semen analysis used during infertility evaluation was assessed for various routine parameters.

The great majority of the patient base were white collar workers with an average age of 39 and a normal- overweight BMI. Smoking rates are very high in this population of 51%. These matters were considered in the analyses.

The principle conclusion was that there was no clear association between parameters of the built-up environment and semen quality other than for the NDVI and sperm vitality. There was also some relationship between proximity to fresh water and semen quality, and of sperm vitality and distance from major roadways. All these data point to some association between the relative ‘greenness of the environment’ and semen quailty amongst men with infertility.

The authors rightly acknowledge the limitations of the study and the mechanism by which any effects may occur remains obscure and a matter of speculation. This study is intriguing and in broad terms points toward the need to understand the environment and lifestyle impact on semen quality and to take measures to ensure that the environment and lifestyle factors are optimised to promote fertility. In the Korean population, the attention clearly needs to be paid to the extraordinary smoking rates which would be deleterious to semen quality.

I have a few minor comments:

1. In terms of proximity to water, the question rises to what type of water? They mention rivers and lakes. Can these vary in the quality of the water in those sites, for example, are there any which are heavily polluted and therefore, amongst men proximal to those, might beneficial effects pf water proximity disappear?

2. The data description I think distances can be expressed to the nearest metre, for example, 486.7m can fairly be rounded to 487m for the purposes of this discussion to make the figures easier to read.

3. The effects do not appear to be dose related. Similarly, when they asked about smoking, it was a Yes or No question. Is it possible to consider those who are very heavy smokers as opposed to minimal smokers and see if there might be some interaction which was not taken into account in the existing statistical approach?

4. Line 191 – They talk about motility being highest in the second quartile, distance from roadway and lowest in the fourth quartile. Why could such a relationship be so non-linear, in other words, why is the first quartile lower than the second? Theoretical reasons might underlie that observation.

6. PLOS authors have the option to publish the peer review history of their article (what does this mean?). If published, this will include your full peer review and any attached files.

Reviewer #1: No

---

## [Author Response · Author response to Decision Letter 0]

6 Jul 2021

Reviewers' comments:

Reviewer #1

 The authors report on 5886 men presenting for infertility evaluation in whom various markers of the ‘built-up environment’ were assessed for their impact on semen quality. Specifically, the distance to fresh-water, to the coast and to major roadways, and the neighbourhood ‘greenness index’, described as the Normalized Difference Vegetation Index [NDVI] as assessed from satellite data. A single semen analysis used during infertility evaluation was assessed for various routine parameters.

The great majority of the patient base were white collar workers with an average age of 39 and a normal- overweight BMI. Smoking rates are very high in this population of 51%. These matters were considered in the analyses.

The principle conclusion was that there was no clear association between parameters of the built-up environment and semen quality other than for the NDVI and sperm vitality. There was also some relationship between proximity to fresh water and semen quality, and of sperm vitality and distance from major roadways. All these data point to some association between the relative ‘greenness of the environment’ and semen quailty amongst men with infertility.

The authors rightly acknowledge the limitations of the study and the mechanism by which any effects may occur remains obscure and a matter of speculation. This study is intriguing and in broad terms points toward the need to understand the environment and lifestyle impact on semen quality and to take measures to ensure that the environment and lifestyle factors are optimised to promote fertility. In the Korean population, the attention clearly needs to be paid to the extraordinary smoking rates which would be deleterious to semen quality.

I have a few minor comments:

Comment #1: In terms of proximity to water, the question rises to what type of water? They mention rivers and lakes. Can these vary in the quality of the water in those sites, for example, are there any which are heavily polluted and therefore, amongst men proximal to those, might beneficial effects pf water proximity disappear?

Response #1: We are thankful for the reviewer’s comments. There might have been hazardous effect by water pollution in some areas. According to the study of Mainali et al. (2018), with temporal and spatial variation in different months of each season, water quality of the river basin of Korea exceeded ‘poor’ category up to 15 percent of times between 2012 and 2016. Given the relatively small proportion of water pollution in the area, we believe that the potential misclassification (benefit of proximity to fresh water) would have minimally biased the result. We discussed this issue (line 248-253, p.16)

Comment #2: The data description I think distances can be expressed to the nearest metre, for example, 486.7m can fairly be rounded to 487m for the purposes of this discussion to make the figures easier to read.

Response #2: We revised the Table 1 using rounded numbers for the distances (Table 1).

Comment #3: The effects do not appear to be dose related. Similarly, when they asked about smoking, it was a Yes or No question. Is it possible to consider those who are very heavy smokers as opposed to minimal smokers and see if there might be some interaction which was not taken into account in the existing statistical approach?

Response #3: There might have been an interaction by amount of smoking in the association between built environment and semen quality. As there is no information about the amount of smoking, we mentioned this issue as one of the limitation (line 256, p.17). 

Comment #4: Line 191 – They talk about motility being highest in the second quartile, distance from roadway and lowest in the fourth quartile. Why could such a relationship be so non-linear, in other words, why is the first quartile lower than the second? Theoretical reasons might underlie that observation.

Response #4: The non-linear association of proximity to fresh water or roadway with semen quality in our study may be explained by U-shaped association between built environment and active traveling which is reported in Liu (2021). We added this discussion (line 134-136, p.16)

---

## [Editor Report · Decision Letter 1]

28 Jul 2021

Land use and semen quality: a fertility center cohort study

PONE-D-21-13270R1

Dear Dr. Kim,

We’re pleased to inform you that your manuscript has been judged scientifically suitable for publication and will be formally accepted for publication once it meets all outstanding technical requirements.

Kind regards,

Stefan Schlatt

Academic Editor

PLOS ONE

Additional Editor Comments (optional):

The authors have dealt with all questions of the reviewer. They have added additional comments to the manuscript and explained a few critical points.

---

## [Editor Report · Acceptance letter]

4 Aug 2021

PONE-D-21-13270R1 

Land use and semen quality: a fertility center cohort study 

Dear Dr. Kim:

I'm pleased to inform you that your manuscript has been deemed suitable for publication in PLOS ONE. Congratulations! Your manuscript is now with our production department. 

Kind regards, 

on behalf of

Dr. Stefan Schlatt 

Academic Editor

PLOS ONE